# Overcoming Chemotherapy Resistance in Germ Cell Tumors

**DOI:** 10.3390/biomedicines10050972

**Published:** 2022-04-22

**Authors:** Zuzana Országhová, Katarina Kalavska, Michal Mego, Michal Chovanec

**Affiliations:** 12nd Department of Oncology, Faculty of Medicine, Comenius University and National Cancer Institute, 833 10 Bratislava, Slovakia; zuzana.orszaghova@nou.sk (Z.O.); michal.mego@nou.sk (M.M.); 2Translational Research Unit, Faculty of Medicine, Comenius University and National Cancer Institute, 833 10 Bratislava, Slovakia; katarina.kalavska@nou.sk; 3Department of Molecular Oncology, Cancer Research Institute, Biomedical Research Center, Slovak Academy Sciences, 845 05 Bratislava, Slovakia

**Keywords:** germ cell tumors, testicular cancer, chemoresistance, cisplatin, molecular mechanisms, novel treatments

## Abstract

Testicular germ cell tumors (GCTs) are highly curable malignancies. Excellent survival rates in patients with metastatic disease can be attributed to the exceptional sensitivity of GCTs to cisplatin-based chemotherapy. This hypersensitivity is probably related to alterations in the DNA repair of cisplatin-induced DNA damage, and an excessive apoptotic response. However, chemotherapy fails due to the development of cisplatin resistance in a proportion of patients. The molecular basis of this resistance appears to be multifactorial. Tracking the mechanisms of cisplatin resistance in GCTs, multiple molecules have been identified as potential therapeutic targets. A variety of therapeutic agents have been evaluated in preclinical and clinical studies. These include different chemotherapeutics, targeted therapies, such as tyrosine kinase inhibitors, mTOR inhibitors, PARP inhibitors, CDK inhibitors, and anti-CD30 therapy, as well as immune-checkpoint inhibitors, epigenetic therapy, and others. These therapeutics have been used as single agents or in combination with cisplatin. Some of them have shown promising in vitro activity in overcoming cisplatin resistance, but have not been effective in clinical trials in refractory GCT patients. This review provides a summary of current knowledge about the molecular mechanisms of cisplatin sensitivity and resistance in GCTs and outlines possible therapeutic approaches that seek to overcome this chemoresistance.

## 1. Introduction

Testicular germ cell tumors (GCTs) represent less than 1% of all male tumors, but they are the most common solid malignancy affecting young men between the ages of 20 and 40 years [1]. The global incidence has been increasing over the past several decades. An estimated 9910 new cases of testicular cancer will be diagnosed in the United States in 2022, resulting in approximately 460 deaths [2]. In Europe, the total number of testicular cancer patients is predicted to increase by 13% from 2010 to 2035, with the largest increase predicted in Eastern European countries [3]. GCTs are classified into two main histologic subtypes: seminomas and non-seminomas (including embryonal carcinoma, choriocarcinoma, yolk sac tumor, teratoma and mixed germ cell tumors), which account for 60% and 40% of the cases, respectively [4].

GCTs are highly curable malignancies with a remarkable 5-year survival rate of approximately 95% [5,6]. Such an extraordinary outcome is unique among solid tumors. Advances in multimodal cancer treatment consisting of surgery, chemotherapy and radiotherapy have resulted in high cure rates in patients with metastatic disease. Cisplatin-based chemotherapy is a standard of care for metastatic germ cell tumors due to their exceptional chemosensitivity. The cure rate after first-line chemotherapy is 70–80%. Patients with relapse can achieve a sustained complete remission (CR) with second-line (salvage) treatment: conventional-dose chemotherapy (CR in 20–40% of cases) or high-dose chemotherapy (HDCT) with autologous stem cell transplant (ASCT) (CR in 60% of cases) [7,8,9,10]. However, about 10–15% of primarily advanced and 3–5% of all GCT patients fail the platinum-based chemotherapy and eventually die of their disease [11,12]. The underlying cause of such chemotherapy failure is attributed to the development of cisplatin resistance in GCTs. Patients who progress during or within one month after completing the initial platinum-based chemotherapy, or patients who relapse/progress after the second-line platinum-based treatment, are considered “platinum-refractory” [13]. The treatment of platinum-refractory GCTs remains a challenge for clinicians and researchers. A deeper understanding of the mechanisms underlying the development of chemoresistance in GCTs might help with improving currently used therapies or discovering new treatments. In addition, the wide range of short-term and long-term toxicities of cisplatin-based chemotherapy encourage the search for new and less toxic treatment options.

In this review, we aim to provide an overview of current knowledge about the molecular mechanisms of cisplatin sensitivity and resistance in GCTs. Furthermore, we summarize the published data on therapeutic interventions that seek to overcome this chemoresistance.

## 2. Cisplatin and Chemosensitivity of Germ Cell Tumors

Cisplatin is one of the first metal-based chemotherapeutic drugs used for the treatment of various types of malignancies, including GCTs, lung, head and neck, ovarian, cervical and bladder cancers [14,15].

### 2.1. Mechanism of Cisplatin Cytotoxicity

The uptake of cisplatin into the cell is mediated by passive diffusion and the copper transporter protein CTR1 [16]. The main mechanism of cisplatin anticancer activity is its interaction with DNA. Cisplatin binds to DNA bases in its reactive form (after aquation inside the cell) and forms cisplatin-DNA adducts. It particularly reacts with the N7-sites of purine bases, preferentially guanine. The formation of intrastrand and interstrand DNA crosslinks causes distortions in the DNA, leading to the inhibition of DNA transcription and cell division. The cell cycle is then arrested, providing adequate time for triggering the DNA repair mechanisms. These remove the DNA damage, or else the programmed cell death (apoptosis) is initiated [15,17]. Another mechanism of cisplatin cytotoxicity is oxidative stress. The production of reactive oxygen species (ROS) that are responsible for DNA damage and mitochondrial dysfunction subsequently leads to apoptosis [18,19]. Cisplatin may induce apoptosis also through the disruption of the plasma membrane, activating the Fas death receptor pathway [20].

The DNA damage response (DDR) is a complex of processes that include the detection of DNA lesions, followed by the initiation of multiple signaling pathways leading to cell cycle arrest (checkpoint activation) and DNA repair in order to preserve genomic integrity [21]. DNA repair mechanisms play an important role in the cytotoxicity of cisplatin. The DNA repair systems involved in removing cisplatin-induced DNA damage include: nucleotide excision repair (NER), mismatch repair (MMR), homologous recombination (HR) and nonhomologous end joining (NHEJ). The NER system recognizes cisplatin-DNA intrastrand crosslinks (ICLs) and excises DNA sequences containing the lesion (of up to 30 base pairs). DNA polymerase subsequently fills the remaining gaps using the intact strand as a template [22,23]. The MMR system replaces the correct base on the non-damaged strand opposite to the cisplatin adduct; however, it is unable to completely repair these DNA lesions, leaving the source of the mismatch intact. This cycle of repetitive cutting and mending of the DNA strand can lead to the formation of double-strand breaks (DSB), which are repaired by HR, NHEJ, and the activation of DNA damage signaling factors [24]. The DDR is regulated by three related protein kinases: ATM (ataxia telangiectasia mutated), ATR (ATM and RAD3-related) and DNA-PKcs (DNA-dependent protein kinase catalytic subunit), as well as by two downstream kinases, checkpoint kinases 1 and 2 (CHEK1 and CHEK2). As a result of DNA damage, they activate tumor suppressor protein p53, which is essential for cisplatin-induced apoptosis [25,26]. In addition, ATM also phosphorylates Mouse Double Minute 2 Homolog (MDM2), which may be a key step in p53 stabilization. MDM2 is an E3 ubiquitin ligase and a physiological p53 antagonist that promotes p53 degradation by forming the MDM2-p53 complex. The ATM phosphorylation of MDM2 reduces the ability of MDM2 to poly-ubiquitinate p53, leading to its stabilization [27]. The use of the MDM2 inhibitor Nutlin-3 resulted in the hyperactivation of p53 pro-apoptotic function in GCT cell lines resistant to cisplatin [28]. The primary mechanism of p53 to induce apoptosis is mediated through the transcriptional activation and repression of target genes. These genes may then activate the apoptotic process via multiple pathways. p53 activates genes encoding p21 and GADD45 (growth arrest and DNA-damage inducible 45) proteins, that lead to cell cycle arrest and the induction of DNA repair. p53 also activates pro-apoptotic genes such as *PUMA*, *NOXA*, caspases, and the MAPK protein family, interacts with Bcl2 family proteins in mitochondria or the cytosol, and facilitates the Fas/FasL pathway, leading to apoptosis [15,29]. 

### 2.2. Chemosensitivity of Germ Cell Tumors

The excellent cisplatin sensitivity of GCTs might be related to their two characteristics: the insufficient DNA repair of cisplatin-induced DNA damage and a hypersensitive apoptotic response [30]. However, more molecular mechanisms involved in GCT chemosensitivity could be identified in future research.

Several proteins, such as high-mobility-group (HMG) proteins, are capable of recognizing and binding to cisplatin-DNA adducts. HMG proteins hence provide shielding and protection from NER-pathway-mediated repair. High-mobility-group box protein 4 (HMGB4) is preferentially expressed in testes. HMGB4 uniquely blocks the excision repair of the cisplatin-DNA adducts, potentiating the sensitivity of GCTs to cisplatin therapy. Knockout of the *HMGB4* gene in a testicular human embryonic carcinoma by CRISPR/Cas9-mediated gene editing induced resistance to cisplatin [31]. In addition, some essential proteins involved in NER, including ERCC1, XPF and XPA, were shown to have low expression levels in GCTs [32]. The overexpression of ERCC1-XPF resulted in decreased cisplatin sensitivity in GCT cell lines. This suggests that ERCC1 and XPF are rate-limiting for the repair of cisplatin-DNA crosslinks in GCTs and contribute to their high sensitivity to cisplatin therapy [33]. Furthermore, reduced homologous recombination in the GCT cell lines, resulting in defective DSB repair, was also correlated with their high cisplatin sensitivity. Poly (ADP-ribose) polymerase (PARP), involved in the repair of DNA single-strand breaks via the base excision repair (BER) pathway, is overexpressed in GCTs compared to normal testes [34]. PARP inhibition was shown to enhance the response of resistant embryonal carcinoma cells to cisplatin by reducing their ability to repair the damage [35].

The hypersensitivity of GCTs to cisplatin is mediated by the rapid induction of multiple apoptotic pathways, with wild type (WT) p53 playing a major role. Mutations in *TP53* have rarely been observed. The incidence of alterations in the *TP53* and *MDM2* genes was higher in chemo-resistant tumors compared to chemo-sensitive ones, but still only in a small subgroup of cisplatin-resistant GCTs [36]. Cisplatin exposure leads to an increased expression of the FAS death receptor, a transcriptional target of p53, and then to the activation of the extrinsic apoptosis pathway through the interaction between FAS and the FAS ligand [28]. The pro-apoptotic proteins PUMA and NOXA, which are involved in the regulation of the intrinsic apoptosis pathway, are both upregulated upon cisplatin treatment [37]. In addition, the hypersensitive apoptotic response in embryonal carcinoma cells is determined by the high expression of Oct-4 (octamer-binding transcription factor 4, also known as POU5F1), a key regulator of pluripotency. High levels of Oct-4 are strongly correlated with a high expression of NOXA, while knockdown of Oct-4 resulted in the reduction of NOXA and decreased cisplatin sensitivity [38]. Interestingly, tissue microarray analyses of GCTs revealed that high NOXA expression was correlated with a good clinical prognosis in patients with embryonal carcinoma [39]. Furthermore, a crosstalk between the extrinsic and the intrinsic apoptosis pathways may strengthen the apoptotic response, and GCTs with WT p53 have higher apoptotic potential caused by increased mitochondrial priming [40]. p53 transactivation of the *CDKN1A* gene, encoding the cyclin-dependent kinase (CDK) inhibitor p21, is essential for the induction of cell cycle arrest in the G1 phase. Oct-4 is also involved in the regulation of p21 expression. Several studies showed that p21 inhibited apoptosis and was highly expressed in more differentiated GCTs, such as mature teratomas, which are resistant to cisplatin-based chemotherapy [41,42]. On the contrary, low levels of p21 were detected in seminomas and embryonal carcinomas, which are primarily sensitive to cisplatin [43]. High cytoplasmic p21 expression has been shown to be one of the key determinants of cisplatin-resistance in embryonal carcinoma, because it protects cancer cells from cisplatin-induced apoptosis. Additionally, high cytoplasmic p21 expression and cisplatin resistance were negatively associated with the expression of Oct4 and miRNA-106b family members [28,44]. Another micro-RNA, a miRNA-302a, is overexpressed in GCTs and its expression increases upon cisplatin treatment in a dose- and time-dependent manner. The upregulation of miRNA-302a results in a higher sensitivity of GCT cells to cisplatin by enhancing G2/M phase arrest and the subsequent progression to apoptosis. It also intensifies the cytotoxic effect of cisplatin by lowering the apoptotic threshold and downregulating p21 [45,46].

Epigenetics is a variety of processes that alter gene expression without altering the sequence of the DNA. This leads to a change in phenotype without a change in genotype. It includes covalent modifications of DNA (e.g., DNA methylation), histone variants, post-translational modifications and non-coding RNAs (ncRNAs). DNA methylation is represented by the addition of methyl groups to the fifth carbon of cytosines, which most commonly occurs in so-called CpG islands [47]. This process is mediated by three different DNA methyltransferases—DNA methyltransferase 1 (DNMT1) maintains methylation patterns during mitosis, while de novo methylation is promoted by DNMT3A and DNMT3B [48]. DNA methylation is one of the most studied epigenetic mechanisms, and it plays an important role in the chemosensitivity of GCTs. The level of DNA methylation appears to be associated with the differentiation status of GCTs and correlates with cisplatin sensitivity in vitro. Germ cell neoplasia in situ (GCNIS) and seminomas are hypomethylated, while more differentiated GCTs have increasing levels of methylation (embryonal carcinomas show an intermediate methylation pattern, yolk sac tumors, choriocarcinomas and teratomas show the highest methylation levels). Low levels of DNA methylation are associated with a higher sensitivity to cisplatin and vice versa [49,50,51,52]. Interestingly, Costa et al. evaluated the differential promoter methylation patterns of five gene promoters (*CRIPTO, HOXA9, MGMT, RASSF1A* and *SCGB3A1*) by which it was possible to distinguish between seminomas and non-seminomas, different non-seminomatous subtypes and even between pure and mixed types. In addition, methylation patterns might also help in the identification of more clinically aggressive tumors [53]. Singh et al. suggested that DNA hypomethylation is a key driver of cisplatin hypersensitivity through at least two mechanisms. First, the more open, pluripotent-like chromatin of GCTs may be more accessible for cisplatin binding and DNA adduct formation. Second, this kind of chromatin may have an inherent transcriptional plasticity, which allows for a stronger acute DNA damage response [54]. 

### 2.3. Summary

The cytotoxicity of cisplatin is mediated by its interaction with DNA and the formation of cisplatin-DNA adducts. This is followed by a DNA damage response and the activation of p53 and signaling pathways leading to apoptosis. The exceptional sensitivity of GCTs to cisplatin appears to be related to the inability of DNA repair systems to properly repair cisplatin-induced DNA damage and to the rapid pro-apoptotic response. In addition, DNA hypomethylation may be a key driver of cisplatin hypersensitivity. 

## 3. Mechanisms of Cisplatin Resistance in Germ Cell Tumors

Multiple factors that contribute to the cisplatin resistance of GCTs on different cellular levels have been identified so far. The mechanisms of cisplatin resistance in general have been classified by some authors according to the sequence of processes that follow the introduction of the drug into the human body: pre-target, on-target and post-target mechanisms. Pre-target resistance includes alterations occurring before cisplatin binds to DNA in the cell, on-target resistance refers to alterations directly related to DNA-cisplatin adducts, and post-target resistance involves alterations in downstream signaling pathways of cisplatin-mediated DNA damage that lead to apoptosis [54,55,56,57]. Although the mechanisms within cancer cells are still the main subject of research, increasing evidence has demonstrated that the tumor microenvironment and immune cells are also important in the development of cisplatin resistance [56]. The possible mechanisms involved in cisplatin resistance are summarized in Figure 1.

### 3.1. Pre-Target Mechanisms

Cancer cells can avoid the cytotoxic potential of cisplatin before it binds to DNA by at least two main mechanisms: decreased intracellular accumulation of cisplatin and increased cisplatin detoxification by cytoplasmic scavengers [55]. Although these mechanisms appear to be an important part of cisplatin resistance in multiple malignancies, they have not been recognized as significant sources of cisplatin resistance in GCTs. However, they may play a contributory role in GCT chemoresistance [54].

The reduced uptake of cisplatin into the cells may be mediated by the downregulation of CTR1, a transmembrane transporter for cisplatin. Studies have shown that CTR1 −/− murine embryonic fibroblasts accumulated much less cisplatin than CTR1 +/+ cells, and were more resistant to even higher concentrations of cisplatin. Furthermore, re-expression of CTR1 in the CTR1 −/− cells restored both cisplatin uptake and cytotoxicity [58,59]. Increased efflux of cisplatin is mostly regulated by the copper-transporting ATPases ATP7A and ATP7B. These transport proteins were shown to be upregulated in cisplatin-resistant cancer cell lines [60]. Studies on lung and ovarian cancer patients reported that high expression levels of ATP7A and ATP7B was correlated with a poor response to cisplatin treatment [61,62].

Resistance to cisplatin is associated with increased levels of cytoplasmic scavenger proteins, such as glutathione (GSH) and metallothionein (MT), that detoxify cisplatin through conjugation, limiting the amount of reactive cisplatin in the cell. The level of GSH appeared to be lower in GCTs compared to some other cancer types and elevated in cisplatin-resistant GCT cell lines [63]. Interestingly, one study showed high levels of a glutathione S-transferase, an enzyme responsible for the conjugation between cisplatin and GSH, in resistant teratoma [64]. Multidrug-resistance-associated proteins (MRPs) and members of ATP-binding cassette (ABC) transporters can also mediate resistance to cisplatin. MRPs are responsible for the removal of cisplatin-GSH conjugates in an ATP-dependent manner. MRP2 is mostly associated with cisplatin resistance and its high expression levels correlated with poor cisplatin response in several types of cancer [65,66].

### 3.2. On-Target Mechanisms

As mentioned before, the exceptional sensitivity of GCTs to cisplatin can be attributed to reduced capacity of repair systems towards cisplatin-induced DNA damage. Consequently, cisplatin resistance may occur due to an increased ability to repair DNA damage or an acquired ability to tolerate unrepaired DNA lesions. 

One of the proposed mechanisms of cisplatin resistance may be an impaired shielding of DNA lesions from repair systems. Awuah et al. demonstrated that the knockout of HMGB4, a protein that protects cisplatin-DNA adducts from repairing through the NER system, induced resistance to cisplatin in GCT cells [31]. Another possible mechanism of GCT chemoresistance is the upregulation of NER pathway proteins, leading to a greater capacity to repair cisplatin-DNA intrastrand crosslinks. The overexpression of ERCC1 and XPF increased the repair of ICLs in GCT cell lines and rendered them more resistant to cisplatin [33]. In addition, the expression of ERCC1 and XPF proteins was shown to be higher in non-seminomas compared to seminomas and normal testis tissue. However, there was no correlation between ERCC1 and XPF expression levels on the one hand and tumor size and TNM stage on the other [67]. High ERCC1 expression was identified in resistant GCT cell lines, as well as in patients with resistant non-seminomas, but no significant association with overall survival was reported [68]. Interestingly, Cierna et al. evaluated the expression levels of NER factors in GCT patients (*n* = 207) and cell lines, and demonstrated the prognostic value of XPA expression on overall survival. This study showed that patients with low XPA expression had significantly better overall survival than patients with high expression (HR = 0.38, 95% CI: 0.12–1.23, *p* = 0.0228). Increased XPA expression was correlated with poor prognostic features in GCTs: non-seminomatous histology, high serum tumor markers, the presence of lung and non-pulmonary visceral metastases, and poor risk group according to the IGCCCG (International Germ Cell Cancer Collaborative Group) classification. In addition, XPA expression was higher in cisplatin-resistant GCT cell lines compared to sensitive ones [69]. 

Several studies have reported a correlation between defective MMR, microsatellite instability (MSI), *BRAF* mutations and cisplatin treatment failure in GCT patients [70,71,72,73]. Honecker et al. evaluated MMR proteins, MSI and *BRAF* mutations in 35 cisplatin-resistant GCTs compared to 100 control GCTs. Resistant tumors showed a higher incidence of MSI, as well as more *BRAF* mutations than controls (26% vs. 0% and 26% vs. 1%, respectively; *p* < 0.0001). In addition, MSI and mutated were *BRAF* correlated with reduced expression of MLH1 (*p* = 0.017 and *p* = 0.008, respectively) [72]. The decreased expression of *MMR* genes, especially *MLH1* and *MSH2*, was also associated with a reduced cisplatin sensitivity of GCT cell lines [73]. The precise molecular mechanism in GCTs is still unclear. However, according to the accepted viewpoints, the MMR system can detect (but not repair) cisplatin-induced DNA lesions, resulting in pro-apoptotic signals. Furthermore, the lack of functional MMR proteins may lead to an increased level of translesional synthesis (TLS) and bypassing of cisplatin-DNA adducts, thus avoiding apoptosis [74,75]. TLS is a DNA damage tolerance mechanism that allows DNA replication to continue beyond cisplatin adducts. This replicative bypass is mediated by a specific group of DNA polymerases including POLH, REV1, REV3 and REV7 [76]. From the perspective of the cancer treatment of multiple malignancies, TLS increases the tolerance of cancer cells to cisplatin-induced DNA damage and allows cancer cells to survive, leading to chemoresistance [77]. 

### 3.3. Post-Target Mechanisms

Alterations in signal the transduction pathways that mediate apoptosis in response to DNA damage represent an essential factor in the cisplatin resistance of GCTs. Decreased expression or dysfunction of pro-apoptotic factors, as well as overexpression of anti-apoptotic factors, can lead to an altered induction of apoptosis. However, the key role of p53 mutations in GCT chemoresistance remains controversial, since mutated p53 occurs only in a subset of refractory GCTs [36,78]. Therefore, modifications of other components regulating the p53 pathway could be more important in the cisplatin resistance of GCTs. Interestingly, di Pietro et al. reported higher expression of p53, MDM2 and p21 in the intrinsic cisplatin-resistant GCT cell lines prior to cisplatin treatment compared to the cisplatin-sensitive GCT cell line and the cell subline with acquired cisplatin resistance. After cisplatin exposure, the levels of p53, MDM2 and p21 increased much more in intrinsic resistant GCT cells in comparison to sensitive GCT cells. The downregulation of p53 made cisplatin-sensitive GCT cells partially resistant to cisplatin-induced apoptosis. By contrast, p53 downregulation sensitized the intrinsically cisplatin-resistant GCT cell lines to cisplatin-induced apoptosis. These findings indicate that p53 transactivation and cisplatin-induced apoptosis in GCT cell lines depend on the cellular context. p53 appears to have a proapoptotic function in cisplatin-sensitive cells and a protective role against apoptosis in intrinsically resistant cells [79]. 

The PDGFR/PI3K/AKT pathway may also play a particular role in the development of cisplatin resistance. The first evidence of a dysregulation of this pathway was reported by Di Vizio et al., who showed that the tumor suppressor gene *PTEN* (regulating this pathway) was extensively expressed in germ cells and GCNIS, while it was practically absent from 56% of seminomas (18/32), 86% of embryonal carcinomas (19/22) and all teratomas (6/6) [80]. Later studies identified a specific role of this pathway in cisplatin resistance, in both cell lines and GCT patients. The overactivation of AKT was observed in cisplatin-resistant cells due to increased mRNA and protein levels for platelet-derived growth factor receptor b (PDGFRb) and PDGF-b ligand. Subsequently, the hyperactivation of the PDGFR/PI3K/AKT pathway resulted in an increased phosphorylation of p21 (leading to its cytoplasmic accumulation) and MDM2 (leading to the inhibition of p53-mediated apoptosis) [81,82]. Furthermore, somatic mutations in AKT1 and PIK3CA were reported exclusively in cisplatin-resistant GCTs [83]. The upregulation of insulin growth factor receptor-1 (IGF1R) expression and signaling has also been found to contribute to acquired cisplatin resistance in an in vitro non-seminoma model. IGF1R was identified as highly expressed and activated in the GCT model cell lines of non-seminoma, with the highest expression in the acquired cisplatin-resistant cell line. In addition, the silencing of IGFR1 led to the apoptosis of resistant non-seminoma cells through their re-sensitization to cisplatin [84]. The deregulation of cyclin D1 (encoded by the *CCND1* gene), a cell cycle regulator, has been described as another potential cause of cisplatin resistance. The overall expression of *CCND1* was higher in cisplatin-resistant cases compared to sensitive samples (*p* < 0.0001), with no significant differences between seminomas and non-seminomas [85]. 

There is increasing evidence that the cisplatin-resistant phenotype can also be maintained (if not entirely generated) by changes in signaling pathways not directly related to cisplatin, a so called “off-target” resistance. This includes the role of cellular differentiation, epigenetic mechanisms, tumor microenvironment and cancer stem cells [55]. 

### 3.4. The Role of Cellular Differentiation

Cellular differentiation is accompanied by epigenetic alterations that increase cisplatin resistance in GCT cells [86]. Chemoresistance is associated with more differentiated histologic subtypes of GCTs, such as teratoma. Therefore, a differentiation-inducing agent all-trans retinoic acid (ATRA) has been used to study the role of differentiation in the chemoresistance of GCT cell lines [87,88,89,90]. Gutekunst et al. demonstrated that the short-term differentiation of embryonal carcinoma cells by ATRA led to the downregulation of NOXA and PUMA, thus inhibiting apoptosis and causing a loss of cisplatin hypersensitivity [37]. The differentiation is accompanied by loss of Oct-4 expression, which has also been associated with increased resistance to cisplatin, especially due to the downregulation of *NOXA* and *PUMA*, as well as the downregulation of miR-17/-106b family members that potentiate activation of p21 [38,82,91]. Other events, such as hypoxia and cisplatin treatment, can also induce the downregulation of Oct-4 and negatively affect cisplatin sensitivity [44,92]. Interestingly, cisplatin treatment selectively depleted Oct4-positive cancer stem cells in a mouse model of metastatic GCT [93]. Taylor-Weiner et al. described the loss of pluripotency markers NANOG and Oct-4 in tumors obtained from cisplatin-resistant metastases [40]. Furthermore, in mouse models of cisplatin-sensitive and -resistant non-seminomatous GCTs, xenografts derived from cisplatin-resistant cell lines exhibited cell areas with embryonal carcinoma morphology, but no Oct-4 expression [94]. 

However, choriocarcinomas and yolk sac tumors, which both lack Oct-4 expression, still respond to cisplatin-based treatment. In addition, one of the first large immunohistochemical analyses of GCTs did not observe a correlation between the expression of Oct-4 and the treatment response [95]. The depletion of Oct-4 has also been shown not to alter the transactivation of p53 target genes, despite a significant decrease in cisplatin sensitivity. Therefore, it has been suggested that Oct-4 does not directly modulate p53 activity but provides a cellular environment that increases p53 proapoptotic activity by maintaining higher levels of NOXA. This indicates that NOXA levels are a central determinant of cisplatin response in GCTs, not the Oct-4 expression [38]. As already mentioned, mature teratomas, as the most differentiated subtype of GCT, that are resistant to cisplatin-based chemotherapy highly express p21 and lack Oct-4 expression. By contrast, seminomas and embryonal carcinomas that are predominantly sensitive to cisplatin hardly express p21. The expression of NOXA was also reduced in teratomas compared to seminomas and embryonal carcinomas [82,96]. The localization of p21 in teratomas was found to be primarily nuclear. However, cisplatin-resistant embryonal carcinoma cells showed high cytoplasmic p21 expression. Interestingly, the re-localization of p21 to the nucleus sensitized embryonal carcinoma cell lines to cisplatin [82]. 

Increasing evidence suggests that the WNT/βcatenin (WNTβ) pathway is involved in the pathogenesis and progression of GCTs. Alterations in the WNTβ pathway may contribute to cisplatin resistance [52,97,98]. The WNTβ pathway is typically activated in the early stages of embryogenesis. It also regulates the differentiation of pluripotent cells [99,100]. Furthermore, it has been associated with the process of carcinogenesis and epithelial–mesenchymal transition [101]. Our translational study evaluated the clinical significance of βcatenin in GCTs and found βcatenin expression in specimens from 213 of 247 patients (86.2%). The expression in seminomas was lower compared to all subtypes of non-seminomas (all *p* < 0.0001). A higher expression was associated with high tumor markers (*p* = 0.035), primary mediastinal non-seminoma (*p* = 0.035) and intermediate/poor risk disease (*p* = 0.033) [102]. Another study reported increased expression levels of *βcatenin* and *CCND1* in two of four cisplatin-resistant GCT cell lines and decreased levels in one embryonal carcinoma cell line. The WNT signaling inhibitor PRI-724 was not significantly effective in cell lines with increased *βcatenin* and *CCND1* expression. However, it showed a higher pro-apoptotic effect in the cell line with decreased expression of *βcatenin* and *CCND1*, probably through the strong activation of caspase-3/7. These findings strongly suggest that the WNT/βcatenin signaling pathway is deregulated in cisplatin-resistant GCTs [103].

### 3.5. The Role of Epigenetic Mechanisms 

DNA methylation as one of the main epigenetic mechanisms is closely associated with the process of tumor differentiation and seems to play a key role in the cisplatin resistance of GCTs [104,105,106,107]. Significant differences in the global DNA methylation pattern of GCTs were observed. Seminomas, which are more undifferentiated tumors that less frequently show cisplatin resistance, are hypomethylated. By contrast, more differentiated non-seminomas show a higher degree of DNA methylation. Embryonal carcinoma with an intermediate level of DNA methylation is also sensitive to cisplatin, but more often shows acquired cisplatin resistance. The highest level of DNA methylation is typical for teratoma, choriocarcinoma and yolk sac tumor, which correlates with cisplatin resistance. However, the hypermethylation of DNA was found in one case of relapsed seminoma after platinum-based chemotherapy. In addition, the demethylation of resistant seminoma cell lines resulted in an increased expression of the cell pluripotency markers NANOG and Oct-4 and a decreased resistance to cisplatin in vitro [50]. Different methylation profiles of several specific gene promoters have been reported in cisplatin-sensitive and -resistant GCTs. Promoter hypermethylation of the *RASSF1A* and *HIC1* genes was observed in resistant non-seminomas, while sensitive non-seminomas showed hypermethylation of *MGMT* and *RARB* [108]. Martinelli et al. showed the association of *CALCA* hypermethylation with non-seminomas (90.5%, 19/21; *p* < 0.026) and cisplatin-refractory disease (47.4%, 09/19; *p* = 0.005). Furthermore, promoter methylation of the *MGMT* and *CALCA* genes correlated with a poor clinical outcome in GCT patients [109]. A growing number of studies have analyzed demethylating agents, their mechanism of action and its effects on GCT cell lines as well as patients. DNA methylation inhibitors are detailed in the section “Treatment approaches to overcome cisplatin resistance in germ cell tumors”.

MicroRNAs (miRNAs) are short single-stranded non-coding RNAs that modulate gene expression at the post-transcriptional level, causing inhibition of mRNA translation or its degradation. They regulate multiple biological processes and have also been linked to carcinogenesis and cancer progression, as well as chemotherapy resistance [110,111]. A study examining almost all known human micro-RNAs reported that 72 of 738 (9.8%) microRNAs were differentially expressed between cisplatin-sensitive and -resistant GCT cell lines. In addition, the miR-371-373 cluster appeared to be involved in cisplatin resistance in GCTs in vitro, since increased levels were found in two out of three resistant cell lines compared to sensitive ones. A possible mechanism might be the inhibition of the p53 pathway. The upregulation of the micro-RNA species hsa-miR-512-3p/-515/-517/-518/-525 and the downregulation of hsa-miR-99a/-100/-145 were also associated with the cisplatin-resistant phenotype in GCTs [112]. 

### 3.6. The Role of Tumor Microenvironment

The chemoresistance of GCTs is a complex and multifactorial phenomenon that appears to be closely related to the tumor microenvironment (TME) [113]. In general, the TME factors affecting cisplatin resistance include: physical components, such as high cell density, fluidic shear stress and extracellular matrix (ECM), which interfere with the delivery and efficacy of cisplatin, and a biological component consisting of biochemical consequences of tumor growth (hypoxia and acidity) and noncancerous cells (e.g., stromal cells, tumor-associated fibroblasts and immune cells) [56]. In addition, GCTs are infiltrated by immune cells that modulate the TME in a variety of ways, including through the secretion of cytokines. The interaction between tumor-infiltrating immune cells and cancer cells creates favorable conditions for tumor survival and growth [113,114]. A study by Siska et al. has shown that seminomas were associated with increased T cell infiltration, as well as PD-L1 expression and PD-1/PD-L1 interaction, but decreased regulatory T cells (Tregs) compared with non-seminomas. However, the advanced disease stage had different immune cell infiltration, irrespective of histological subtype. The T cell and natural killer (NK) cell populations responsible for anti-tumor immunity were decreased, while regulatory T cells (Tregs), neutrophils, mast cells and macrophages, with potentially pro-tumor immune activity, were significantly increased [115]. An intensive crosstalk between the TME and DNA damage and repair pathways has also been reported [116]. A recent study evaluated the interaction between the immune TME and endogenous DNA damage levels in GCTs by the co-cultivation of peripheral blood mononuclear cells (PBMCs) from healthy donors and GCT cell lines. PBMCs co-cultivated with cisplatin-resistant cell lines showed significantly higher DNA damage levels than PBMCs co-cultivated with sensitive cell lines. In addition, endogenous DNA damage levels above the cut-off value were associated with increased numbers of NK-cells, Tregs and CD16-positive dendritic cells [117]. Cancer cells are able to suppress anti-tumor immunity through PD-1/PD-L1 signaling in the TME. PD-L1 expression in specimens from 140 patients with GCTs was significantly higher in comparison with normal testicular tissue (*p* < 0.0001). Choriocarcinomas expressed the highest level of PD-L1, with declining positivity in embryonal carcinoma, teratoma, yolk sac tumor and seminoma. Furthermore, PD-L1 expression was associated with the poor prognostic features of GCTs [118]. The expression of PD-L1 was also evaluated in the tumor-infiltrating lymphocytes (TILs) of tumor samples from 240 patients with GCTs. PD-L1 expressing TILs were more frequently found in seminomas (95.9% of patients) and embryonal carcinomas (91%) compared to yolk sac tumors (60%), choriocarcinomas (54.5%) or teratomas (35.7%) (all *p* < 0.05). In addition, patients with high infiltration of PD-L1-positive TILs had significantly better progression-free survival (PFS) (HR = 0.17, 95% CI 0.09–0.31, *p* = 0.0006) and overall survival (OS) (HR = 0.08, 95% CI 0.04–0.16, *p* = 0.001) in contrast to patients with lower infiltration of TILs [119]. All these findings suggest a potential major role for the TME, especially immune cells, in progression, cisplatin sensitivity and resistance of GCTs. 

### 3.7. The Role of Cancer Stem Cells

Cancer stem cells (CSCs) represent a subpopulation of tumor cells with cancer initiation ability, clonal long-term repopulation potential and self-renewal capability. CSCs are considered to be an origin of cancer and they can switch between stem and non-stem cell state. In addition, CSCs are resistant to conventional chemotherapy and radiation therapy [120,121,122]. Their identification is based on the expression of specific cell surface markers. CSCs have the characteristics of normal stem cells and differentiated cancer cells, and therefore they share both stemness-associated and tissue-specific markers. CD24, CD26, CD44, CD133, CD166, Ep-CAM (also called CD326 or epithelial-specific antigen) and aldehyde dehydrogenase (ALDH) are examples of CSC-specific surface markers [123]. 

ALDH is a NAD(P)-dependent enzyme involved in cellular detoxification and resistance to chemotherapeutic agents by oxidation of cellular aldehydes. In particular, ALDH1 family members (ALDH1A1, ALDH1A2, and ALDH1A3) are responsible for the increased self-renewal, survival and proliferation of CSCs [124]. A high expression of ALDH1 has been associated with chemoresistance and metastasis formation, and it has even been correlated with a poor clinical prognosis in several cancer types [125,126,127,128,129]. The ALDH1A3 marker was significantly overexpressed in all histological subtypes of GCTs compared to normal testicular tissue. In addition, high ALDH1A3 expression and increased ALDH activity were detected in cisplatin-resistant embryonal carcinoma cell lines. However, no association was found between ALDH1A3 expression in tumor cells and tumor primary, IGCCCG risk group, number of metastatic sites or S-stage [130]. 

### 3.8. Summary

The molecular mechanisms responsible for cisplatin resistance in GCTs can be classified as pre-target, on-target and post-target. Pre-target mechanisms include decreased intracellular accumulation of cisplatin—reduced cisplatin uptake by CTR1 and increased efflux of cisplatin by ABC transporters—and increased cisplatin detoxification by cytoplasmic scavengers, such as GSH. On-target mechanisms are mediated by an increased ability to repair DNA damage or an acquired ability to tolerate unrepaired DNA lesions. Post-target mechanisms involve changes in apoptosis signaling pathways, with important roles for p53, MDM2, p21 and other proteins. These processes lead to cell cycle arrest and the inhibition of apoptosis, resulting in chemoresistance. Other “off-target” factors of cisplatin resistance include cellular differentiation, epigenetic mechanisms (especially DNA hypermethylation), cancer stem cells and a tumor micro-environment with a key role for immune cells.

## 4. Treatment Approaches to Overcome Cisplatin Resistance in Germ Cell Tumors

A better understanding of the molecular mechanisms involved in cisplatin sensitivity and resistance in GCTs may help researchers to identify potential therapeutic targets. Numerous preclinical and clinical trials have studied the effect of different treatment approaches in refractory GCTs to overcome chemoresistance. These include single-agent and combinational chemotherapy, but also targeted therapy, immunotherapy, epigenetic therapy, as well as a variety of other therapeutic agents, either alone or in combination with cisplatin. Only some of them have demonstrated the expected efficacy in vitro, in vivo and in patients’ cohorts. However, some therapeutic agents appear to be promising and require further research. Possible treatment approaches in refractory GCTs are discussed below and potential therapeutic agents used in clinical trials are summarized in Table 1.

### 4.1. Different Chemotherapy Regimens

Many conventional chemotherapeutics with different cytotoxicity mechanisms compared with cisplatin have been evaluated in the treatment of refractory GCTs. These studies were usually small, single-arm phase II trials. Single-agent chemotherapy as well as combinational chemotherapy approaches have been used. However, most of the trials failed to induce clinically relevant responses, with only a small number of patients achieving complete or durable remissions. 

Single-agent chemotherapy trials included treatment with gemcitabine [155,156], oxaliplatin [157,158], capecitabine [159], irinotecan [160], paclitaxel [161,162,163,164], ifosfamide [165], oral temozolomide [166] and oral etoposide [167]. There have been several agents with documented efficacy and good tolerability, namely oral etoposide, gemcitabine, oxaliplatin and paclitaxel. However, objective responses were achieved in approximately 10–30% of cases and complete remissions (CR) were rarely observed. Furthermore, Feldman et al. performed seven single-agent phase II trials in refractory GCT patients (*n* = 90) from 1990 to 2008, evaluating the effects of suramin, ATRA, topotecan, pyrazoloacridine, temozolomide, ixabepilone and sunitinib. An objective response was achieved in one of 90 patients (1%), which was a partial response to ixabepilone. Fifteen patients (17%) achieved a stable disease—three patients each on sunitinib, ixabepilone, topotecan and temozolomide, two patients on ATRA and one patient on suramin. However, the majority of the patients (*n* = 74; 82%) had progressive disease. The median PFS and OS were 1.0 month (95% CI, 0.8–1.3) and 4.7 months (95% CI, 3.5–6.4), respectively [168]. 

Oral etoposide at a dose of 50 mg/m2 daily appeared to be a well-established option for patients with refractory GCT. In a small phase II trial, 6 out of 21 evaluable patients (28%) responded with a more than 90% decrease in tumor markers and three of them were accompanied by a more than 50% regression in radiologically measurable disease. The most common side effects were of hematologic origin (particularly anemia and granulocytopenia), with eight patients requiring temporary cessation of treatment and four patients requiring dose reductions. The median duration of treatment was 11.5 weeks (range = 2 to 30 weeks), with six patients still continuing the treatment at the end follow-up period [167]. 

Cabazitaxel, a novel taxane developed to overcome resistance to docetaxel and paclitaxel, has shown promising preclinical activity in GCT cell lines, independent of cisplatin resistance levels [169]. However, it showed limited activity in heavily pre-treated GCT patients [131]. The first clinical case series of cabazitaxel treatment for refractory GCT patients (*n* = 13) reported a 12-week PFS rate of 31%. Objective responses were observed in two patients (15%) and a tumor marker decline ≥ 50% in three patients (23%), accounting for a disease control rate of 39%. Disease progression without any signs of response was reported in seven patients (54%). The median PFS and OS were 7 weeks (95% CI, 4.9–9.1) and 23 weeks (95% CI, 8.6–37.4), respectively. Cabazitaxel was well tolerated, with rare dose reductions (15%) and hematotoxicity as the most common side effect [131]. In addition, two phase II clinical trials (NCT02115165, NCT02478502) are currently underway to prospectively evaluate cabazitaxel in refractory GCT patients. 

Combination chemotherapy regimens have been tested in numerous phase II trials and some of them have demonstrated activity in patients with multiply relapsed and refractory GCTs [170]. A majority of these chemotherapy regimens consisted of a platinum backbone of cisplatin [171,172,173,174,175,176], oxaliplatin [177,178,179,180,181,182,183,184] or nedaplatin [171,185,186], in combination with other agents, including gemcitabine, paclitaxel, ifosfamide, irinotecan or epirubicin. A small number of combination regimens did not use a platinum backbone [187,188,189]. The studies were also mostly small and single-armed. Response rates have been reported to be up to 20–50%, with durable remissions in 5–20% of patients [170]. The combination of gemcitabine, cisplatin and paclitaxel (GCP) showed good efficacy in groups of patients (*n* = 75) after failure of two or three cisplatin-based lines of therapy, including HDCT. Forty-two patients (56%) had cisplatin-refractory disease. The study reported CR in eight patients (10.7%) and partial remission with negative markers (PRm-) in 29 patients (38.7%), resulting in an overall response rate of 49.3% (95% CI, 37.6%–61.1%). Thirty-three patients (44%) underwent surgery after treatment with GCP, which was radical in 14 cases (42.4%). The median PFS and OS were 5 months and 13 months, respectively. Two-year PFS and OS in cisplatin-refractory patients were 7.1% (95% CI, 2.4%–21.3%) and 22.0% (95% CI, 11.8–40.7%), respectively. The toxicity of this regimen was primarily hematologic, with 40 patients (53.3%) experiencing grade 3 or 4 myelosuppression, but low rates of grade 3 or 4 ototoxicity and neurotoxicity were reported [176]. The combinations of gemcitabine and oxaliplatin with or without paclitaxel (GEM/OX or GOP) have been the most studied chemotherapy regimens in refractory GCT patients [179,180,181,182,183,184]. A German triple combination phase II trial of GOP in patients with cisplatin-refractory or multiply relapsed GCTs (*n* = 41) showed an overall response rate of 51% (95% CI, 35–67%). CR was observed in 5% of patients (*n* = 2), PRm- in 34% and PR with positive tumor markers (PRm+) in 12% of patients. In addition, remissions lasting at least 2 years were achieved in 17% of patients (*n* = 7), some of which underwent complete resection of residual masses [179,181]. These findings also emphasize the important role of secondary surgery in this setting. 

In conclusion, combination chemotherapy appears to provide higher response rates compared to single-agent chemotherapy. Therefore, preferably a triple combination followed by subsequent radical secondary resection might be an effective option in patients with cisplatin-refractory or multiply relapsed GCTs. However, in patients who are not eligible for the combination chemotherapy due to comorbidities, performance status or the toxicities of previous treatment, the use of single-agent chemotherapy may be considered. 

### 4.2. Targeted Therapy

Targeted therapies are used to treat a variety of solid tumors in a selected population who may benefit from these agents. The use of targeted agents in the treatment of GCTs has not yet found a stable position and remains questionable due to the low percentage of patients who relapse and the population diversity, including several histological subtypes. However, several targeted agents showed promising in vitro activity and have therefore already been tested alone or in combination in small, non-randomized phase I/II studies. In addition, the available data revealed a possible outcome of some targeted treatments in a selected group of patients with cisplatin-resistant GCTs.

#### 4.2.1. Tyrosine Kinase Inhibitors (TKIs)

Receptor tyrosine kinases (RTKs) play an important role in the development and progression of GCTs. In particular, several classes have been studied in GCTs, namely, vascular endothelial growth factor receptor (VEGFR), platelet-derived growth factor (PDGF), hepatocyte growth factor receptor (HGFR), epidermal growth factor receptor (EGFR) and stem cell growth factor receptor (SCFR). 

GCTs overexpress VEGF, PDGF and their respective receptors VEGFR-2 and PDGFR-β compared to normal testicular tissue, leading to a strong activation of the AKT signaling pathway in cisplatin-resistant GCT cell lines [81]. The multikinase inhibitors sunitinib and pazopanib were able to re-sensitize resistant GCT cells to cisplatin in vitro [81,133,190]. Furthermore, pazopanib as monotherapy or combined with lapatinib showed activity in murine models of cisplatin-sensitive and cisplatin-resistant GCTs [191]. In a phase II clinical trial, the use of sunitinib as a single agent in heavily pretreated refractory GCT patients (*n* = 32) resulted in a total response rate of 13% (only partial remissions). The median PFS was 2 months (95% CI, 1.4–2.6), with PFS at 6 months of 11%. The median OS was 3.8 months (95% CI, 3–6.6), with an OS at 6 months of 36.4% [133]. Another phase II clinical trial reported an exceptional response to sunitinib in one of five heavily pre-treated patients with refractory GCTs, with the response was maintained during 17 months [135]. A Slovak study also concluded that sunitinib had limited efficacy in refractory GCT patients (*n* = 10), with two PRs after two treatment cycles, followed by progression after four cycles. In a different study on 10 patients with refractory GCTs, sunitinib only achieved tumor marker stabilization as the best response, with all the patients progressing during the first three cycles [132]. Pazopanib in a small phase II trial (*n* = 43) only induced the reduction of tumor markers in 70.3% of patients, two partial responses (4.7%) and 19 cases of stable disease. The median follow-up period was 29.6 months (range = 10.6–35.8 months). The 3-month PFS was 12.8% (95% CI, 5.7–28.9%) and the 24-month OS was 14.2% (95% CI, 6.0–33.7%) [136]. Sorafenib in monotherapy also did not demonstrate significant activity in patients with highly refractory GCTs (*n* = 18), inducing only tumor marker decline in eight patients and disease stabilization for more than 12 months in three patients [137]. Another multikinase inhibitor nintedanib induced apoptosis and growth arrest in both cisplatin-sensitive and cisplatin-resistant GCT cell lines in vitro [192]. However, a clinical trial in GCTs has not yet been initiated. Interestingly, a study group from Indiana University is currently investigating the use of cabozantinib for patients with refractory GCTs in a phase II clinical trial (NCT04876456). In addition, two different antiangiogenic agents targeting VEGFR-2, HP-2 and HP-14, have been tested preclinically in cisplatin-sensitive and -resistant cell lines. HP-14 effectively inhibited the proliferation of cisplatin-resistant GCT cells and re-sensitized them to cisplatin treatment, but further clinical studies are needed [193]. 

SCFR (also named c-KIT or CD117) is also overexpressed in GCTs, predominantly in seminomas, but only in a small percentage of non-seminomas, and almost absent in choriocarcinomas and teratocarcinomas [194,195,196]. Mutations of c-KIT have been found in 10–40% of GCTs and more frequently included exon 17 than exon 11. In one European study, mutated KIT was significantly associated with bilateral testicular GCTs [197], but it was not confirmed in a Japanese study [198]. The potential role of imatinib, a c-KIT inhibitor, has been investigated in GCTs, but several results have shown low clinical efficacy, probably due to exon 17 mutations responsible for resistance to KIT-targeted therapy [138,199]. However, two clinical cases with an objective response to imatinib have been reported in patients with c-KIT positive refractory germ cell tumors, one in monotherapy and the other in combination with a GOP chemotherapy regimen [200,201]. 

HGFR (also named MET) is dysregulated in many solid tumors and activates the MAPK and PI3K-AKT signaling pathways. Despite preclinical activity, a phase II trial of tivantinib, a selective small-molecule MET inhibitor, did not demonstrate single-agent activity in patients with relapsed and refractory GCTs (*n* = 27), with only five patients achieving a stable disease. The median PFS was 1 month (95% CI, 1–2 months) and the median OS was 6 months (95% CI, 3–8 months) [139].

#### 4.2.2. mTOR Inhibitors

The loss of the tumor suppressor gene *PTEN*, which is usually highly expressed in germ cells, marks the transition from GCNIS to invasive GCT [80]. The inactivation of PTEN is linked to the deregulation of the PI3K/AKT pathway and the increase in the mammalian target of rapamycin (mTOR). Overactive mTOR enhances cell growth, cell cycle progression, proliferation and survival. In particular, seminomas were found to express mTOR pathway proteins extensively in an immunohistochemical study [202]. Two phase II clinical trials tested the mTOR inhibitor everolimus in refractory or multiply relapsed GCTs, showing limited efficacy in this group of patients [140,141]. The first clinical study (*n* = 15) reported no objective response, but six patients (40.0%) achieved 12-week PFS and one patient achieved prolonged disease stabilization for 22.2 months. The median PFS was 1.7 months (95% CI, 1.1–4.0 months) and the median OS was 3.6 months (95% CI, 2.0–11.0 months) [140]. The second study (*n* = 25) also did not achieve any objective response, with only one patient showing a stable disease after 6 weeks of treatment. The median PFS and OS were 7.4 weeks (80% CI, 4.9–7.6 weeks) and 8.3 weeks (80% CI, 7.1–9.1 weeks), respectively [141]. A small clinical trial evaluated the combination of sirolimus plus erlotinib (NCT01962896) to assess the effects of combined approaches targeting both EGFR and mTOR. However, this trial was terminated due to low accrual. 

#### 4.2.3. PARP Inhibitors

The PARP family of proteins, which is involved in the BER system of DNA repair, is overexpressed in GCTs compared to normal testicular tissue. The overexpression of PARP appears to be an early event in GCT development. Patients with low PARP expression in the primary tumor had better OS than patients with high PARP expression, but the difference was not statistically significant (5-year OS 89.2% vs. 78.7%; HR = 0.50; 95% CI, 0.21 to 1.17; *p* = 0.12) [34]. In addition, the inactivation of PTEN is also associated with defects in the HR DNA repair pathway and an improved response to PARP inhibitors [203]. Two phase II clinical trials evaluated the use of PARP inhibitors in refractory or multiply relapsed GCT patients, the first in monotherapy [142] and the second in combination with cytotoxic agents [143]. Olaparib as a single agent showed marginal activity in a group of heavily pretreated GCT patients (*n* = 18) with no partial responses, with five patients (27.8%) achieving stable disease, and progressive disease in 13 patients (72.2%). The 12-week PFS was 27.8% (95% CI, 10.1–48.9%) and the 12-month OS was 27.8% (95% CI, 10.1–48.9%) [142]. Another study evaluated veliparib in combination with gemcitabine and carboplatin (*n* = 15), also showing limited efficacy with four partial remissions (26.7%) and five cases of disease stabilization (33.3%). The 12-month PFS was achieved in only one (6.7%) patient. The median PFS and OS were 3.1 months (95% CI, 2.2–3.9) and 10.5 months (95% CI, 8.9–11.1), respectively [143]. 

#### 4.2.4. CDK Inhibitors

GCTs are characterized by the appearance of an abnormal chromosome, isochromosome 12p. This chromosomal region contains genes involved in G1/S cell cycle checkpoint regulation, namely, cyclin-dependent kinase 4 (CDK4), which activates retinoblastoma protein Rb, and its catalytic partner cyclin D2. It has been shown that GCTs frequently upregulate the expression of CDK4 and cyclin D2 [204]. Undifferentiated GCTs hardly express Rb protein, while more differentiated teratomas have shown high expression levels of Rb protein, which promotes cell growth after phosphorylation by CDK4/6 [205]. Therefore, selective inhibitors of CDK 4/6 have been used in clinical trials by Vaughn et al. to inhibit growing teratomas [144,206]. In a phase I trial, three patients with unresectable growing teratomas received palbociclib. Two patients had a stable disease for 18 and 24 months, respectively, and one patient achieved a partial response for 22 months [206]. In a phase II study in refractory GCT patients with Rb-expressing tumors (*n* = 29), treatment with palbociclib resulted in a 24-week PFS rate of 28% (90% CI, 15–44%) (*n* = 8). In addition, patients with teratoma and teratoma with malignant transformation had significantly better PFS than patients with other histological subtypes [144]. A randomized placebo-controlled phase II trial investigated the CDK4/6 inhibitor ribociclib in GCT patients with unresectable and progressive teratoma without malignant transformation. The trial was prematurely closed due to slow accrual, but the ribociclib group (*n* = 8) showed a 24-month PFS rate of 71% [145]. 

#### 4.2.5. Anti-CD30 Therapy

The CD30 cell surface protein is commonly expressed in activated lymphocytes, Hodgkin’s lymphoma cells and anaplastic large cell lymphoma (ALCL) cells, but it can also be found on GCT cells, especially on embryonal carcinomas and less frequently on pure seminomas and seminomatous components of mixed GCTs [207,208]. Interestingly, CD30-expression changes during cisplatin-based chemotherapy, with only 40–44% of initially CD30-positive GCTs maintaining the positivity after systemic treatment. The maintenance of CD30-expression is a marker of a persistent vital tumor and it is correlated with a poorer prognosis [209,210]. The antibody–drug conjugate brentuximab vedotin specifically targets CD30-expressing cells. This agent showed potent antiproliferative and pro-apoptotic activity in vitro against both CD30-positive and CD30-negative embryonal carcinoma cell lines [211]. In a phase II clinical trial, nine patients with CD30+ refractory GCTs received brentuximab vedotin, with an objective response rate of 22.2% (one CR and one PR). The 3-month PFS was 22.2% (95% CI, 3.4–51.3%) and the 6-month OS was 77.8% (95% CI, 36.5–93.9%). In addition, the immunomodulatory effect of brentuximab vedotin was observed with a decrease in activated T cells, granulocytes and mature dendritic cells, and an increase in immature dendritic cells [146]. In a small series of three GCT patients with refractory CD30+ embryonal carcinoma, all patients obtained a clinical benefit after treatment with brentuximab vedotin (one PR, one serologic response and one stable disease), but for a very short duration of only 2 months [212]. Another clinical trial in five CD30+ patients with relapsed or refractory GCTs of different histologies reported an objective response in two patients, including one durable CR (more than 46 months after the discontinuation of the treatment) and one PR at a single time point [147]. 

### 4.3. Immunotherapy

Immune mechanisms are involved in the pathogenesis and the response to treatment of GCTs. Therapeutic disruption of PD-1/PD-L1 interaction with immune checkpoint inhibitors restores the antitumor immune response and can induce durable remissions even in patients with chemo-resistant solid tumors [213,214]. Two main studies have evaluated the expression of PD-L1 in GCTs and both reported higher expression of PD-L1 in tumors compared to normal testicular tissue [118,215]. In addition, PD-L1 expression has been shown to have a prognostic value in GCTs, as already mentioned. Patients with low expression of PD-L1 in tumor cells had better PFS and OS compared to patients with high PD-L1 expression [118]. However, in tumor-infiltrating lymphocytes, a high expression of PD-L1 was associated with better PFS and OS [119]. All these findings suggest that the PD-1/PD-L1 pathway could be a new therapeutic target in GCTs. However, immune checkpoint inhibitors have not yet demonstrated the expected clinical efficacy. 

In a case series of seven patients with refractory GCTs who had received immunotherapy with either pembrolizumab or nivolumab, four patients progressed rapidly and died shortly after the initiation of treatment. However, three patients received treatment for at least 6 months and long-term remission was achieved for two of them, both with high PD-L1 expression [150]. The first prospective phase II trial evaluating immune checkpoint inhibitors in GCTs included 12 patients with refractory non-seminomas (two of them were PD-L1 positive). The patients were treated with pembrolizumab and no objective response was observed. Only two patients (both PD-L1 negative) achieved radiographically stable disease within 28 and 19 weeks, but with a continuing increase in tumor markers [149]. In another phase II study, 12 patients with refractory GCTs (10 men and 2 women) were treated with pembrolizumab. No objective response was reported, but three patients had radiographically stable disease lasting 10.9, 5.5 and 4.5 months, respectively. The median PFS was 2.4 months (95% CI, 1.5–4.5 months) and the median OS was 10.6 months (95% CI, 4.6–27.1 months) [148]. Interestingly, one Japanese study recently reported the case of a refractory GCT patient with PD-L1 positivity and high MSI rapidly responding to pembrolizumab. In addition, the patient was free from disease progression for 6 months from the start of immunotherapy [216]. Avelumab showed no efficacy in patients with multiple relapsed/refractory non-seminomas (*n* = 8). At a median follow-up of 2.6 months (range = 0.3–14.4), all patients had disease progression and seven patients (87.5%) died. The 12-week PFS was 0%, the median PFS was 0.9 months (95% CI, 0.5–1.9) and the median OS was 2.7 months (95% CI, 1.0–3.3) [152]. An APACHE study assessing durvalumab (anti-PD-L1) with or without tremelimumab (anti-CTLA4) in refractory GCT patients (*n* = 22) showed one case of PR and one case of stable disease with marker decline (both patients from combination arm). On the basis of these results, the monotherapy arm was closed [153]. There are several ongoing clinical trials using immune checkpoint inhibitors in refractory GCTs as single agents or in combination (NCT03158064, NCT02834013 and NCT02832167). 

### 4.4. Epigenetic Therapy

Research into the epigenetic mechanisms of GCTs has demonstrated that DNA hypermethylation and histone deacetylation are associated with cisplatin resistance. Therefore, treatment approaches targeting DNA methyltransferases (DNMTs) and histone deacetylases (HDACs) have been intensively studied in monotherapy and in combinations with each other or other drug classes [217]. Hypomethylating agents (HMAs), such as decitabine, guadecitabine and 5-azacitidine, which inhibit DNMTs, showed promising results in different in vitro and preclinical in vivo studies. Exposure to hypomethylating agents led to the re-expression of tumor suppressor genes, the activation of p53 and a proapoptotic response, and thus to the re-sensitization of GCT cells to cisplatin [50,218,219,220,221]. Interestingly, in a preclinical in vivo study, guadecitabine completely inhibited progression and induced the complete regression of cisplatin-resistant embryonal carcinoma xenografts. This effect seemed to be the consequence of the induction of p53 targets, immune-related pathways and the repression of pluripotency genes [220]. Albany et al. conducted a phase I clinical trial combining guadecitabine with cisplatin in recurrent cisplatin-resistant GCT patients (*n* = 14). The median follow-up was 6.9 months (range = 0.03 to 26.5). Three patients had a partial response. Two of them, including one with primary mediastinal GCT, completed the study and achieved a complete response assessed by serum tumor markers, with a sustained remission of 5 and 13 months and a survival of 16 and 26 months, respectively. The median OS was 7.8 months (95% CI, 2.7–12.5) and the median PFS was 1.7 months (95% CI, 0.9–3.7). The overall response rate was 23% and three patients also achieved a stable disease, indicating a clinical benefit rate of 46%. The most common grade 3–4 adverse events were neutropenia (79%), thrombocytopenia (43%) and anemia (36%) [154]. Histone deacetylase inhibitors (HDACi), such as romidepsin and trichostatin A, also showed preclinical activity in vitro and in vivo. They induced apoptosis and reduced tumor size, the proliferation rate and angiogenesis in GCTs [222,223,224,225]. A combination of epigenetic therapy with immunotherapy might be considered a good treatment alternative due to the known epigenetic regulation of the immune system [226]. 

### 4.5. Other Agents and Potential Therapeutic Targets

ALDH is overexpressed in all histological subtypes of GCTs compared to normal testicular tissue. In addition, high ALDH1A3 expression and increased ALDH activity were detected in cisplatin-resistant embryonal carcinoma cell lines. The ALDH inhibitor disulfiram in combination with cisplatin showed synergy for cisplatin-resistant GCT cell lines and inhibited the growth of resistant xenografts [130]. Since disulfiram augmented cisplatin toxicity in vitro and in vivo, ALDH inhibition appeared suitable for the combination therapy to achieve an antitumor effect in patients with refractory GCTs. A phase II clinical trial of disulfiram and cisplatin combination in refractory GCT patients is still ongoing (NCT03950830), and results will be published in the second quarter of this year.

The Wnt/β-catenin signaling pathway, which appears to be dysregulated in refractory GCTs, may be inhibited by PRI-724. This agent demonstrated a proapoptotic effect in cisplatin-resistant GCTs [103]. The available data suggest that the inhibition of the Wnt/β-catenin pathway might be beneficial in the treatment of refractory GCT patients, but further research is needed. In addition, clinical trials with PRI-724 in GCTs are missing. 

MDM2 has been shown to be a promising target in GCT cell lines. The small-molecule MDM2 inhibitor Nutlin-3 blocks the interaction between p53 and MDM2, thus inducing apoptosis. In addition, synergistic effects on GCT cells were observed after combined Nutlin-3 and cisplatin treatment [28,227]. Several other MDM2 inhibitors, including RG7388, AMG-232 and ALRN-6924, are also under investigation, but no clinical studies have yet been initiated in patients with refractory GCTs. 

### 4.6. Summary

Several treatment approaches to overcome chemoresistance in GCTs have been studied. Preclinical and clinical studies have used various therapeutic agents alone or in combination with cisplatin. Oral etoposide as a single agent and GCP/GOP as combinational chemotherapy have shown some clinical effects in refractory or multiply relapsed patients. The use of targeted therapies such as TKIs, mTOR inhibitors, PARP inhibitors, CDK inhibitors, and anti-CD30 treatment remains questionable in GCTs due to its limited clinical activity in patient cohorts. However, their use may be a reasonable option in selected groups of patients. Immune checkpoint inhibitors have shown very low efficacy in clinical trials. On the contrary, the use of HMAs has shown promising results. In addition, the combination of HMAs with immunotherapy could be considered in the future. These data suggest that the ideal therapeutic agent for refractory GCTs is still absent. However, multiple clinical trials are currently underway.

## 5. Future Directions and Considerations

Cisplatin resistance is an emerging clinical issue. Cisplatin-resistant GCTs are associated with poor prognosis. Most of these young men eventually die of their disease. Our effort to overcome cisplatin resistance should be based on an understanding of the molecular basis of this phenomenon. This can only be achieved through ongoing preclinical and clinical research. 

The pitfalls of clinical research in refractory GCTs are that the cohorts are small and that they consist of heavily pretreated patients. Patients enrolled in clinical trials already suffer from various severe toxicities resulting from cumulative doses of chemotherapeutics, especially cisplatin. This may limit the use of emerging therapeutic agents, which carry their own toxicities. Another issue experienced over recent years is a lack of clinical activity in agents that had shown a promising efficacy in preclinical studies. Despite these disappointing obstacles, clinical studies emerging from high-quality translational research are imperative in the search for new treatment approaches and therapeutic combinations. One approach for conducting successful clinical trials in the rare clinical scenario of GCT treatment resistance is centralized patient care within specialized centers. Ultimately, joining forces in the international consortia for the treatment of refractory GCTs seems to be essential for future advancements. 

Furthermore, the role of subsequent tumor biopsy, liquid biopsy and next-generation sequencing after a relapse or disease progression should become a standard clinical procedure in the search for individual biomarkers within the precision genomics field. Such an approach allows for precision diagnostics in the clonal evolution of the disease in the individual patient with the possibility of discovering a rare targetable molecular biomarker. 

Ongoing preclinical and clinical research is needed to continuously investigate the molecular mechanisms of cisplatin resistance and to identify the potential therapeutic targets and prognostic biomarkers. With strongly established translational research, novel therapeutic agents may be challenged in clinical studies, which may ultimately lead to strategies for overcoming chemotherapy resistance in GCTs. 

## 6. Conclusions

This review summarized the current evidence on cisplatin sensitivity and resistance in GCTs. The molecular mechanisms responsible for chemotherapy resistance and possible therapeutic approaches in refractory GCT patients were discussed. The remarkable sensitivity of GCTs to cisplatin-based chemotherapy is related to alterations in the DNA repair systems of cisplatin-induced DNA damage and the rapid induction of multiple apoptotic pathways, with p53 playing a major role. Multiple molecular mechanisms contribute to cisplatin resistance; they can be classified as pre-target, on-target and post-target. In addition, epigenetic mechanisms, cellular differentiation, as well as cancer stem cells and the tumor microenvironment with immune cells, play an important role in tumor response to treatment. Multiple molecules and molecular pathways have been identified as potential therapeutic targets to overcome cisplatin resistance. Therapeutic agents that have been evaluated in preclinical and clinical studies include different chemotherapeutics, novel targeted therapies (tyrosine kinase inhibitors, mTOR inhibitors, PARP inhibitors, CDK inhibitors, anti-CD30 therapy, immune-checkpoint inhibitors, etc.), epigenetic therapy and others. These therapeutics have been used in monotherapy or in combination with one another or with cisplatin. Some of these agents have shown promising preclinical activity in overcoming cisplatin resistance. However, the current results from clinical trials in patients with refractory GCTs are largely disappointing. While improving outcomes in the palliative treatment of patients with refractory disease is important, our final goal should be cure for all patients with GCTs. Finally, more preclinical and clinical studies are needed to better understand the molecular mechanisms of cisplatin resistance, to identify potential therapeutic targets and prognostic biomarkers, and to evaluate new therapeutic agents that can possibly overcome chemoresistance in GCTs.

## Figures and Tables

**Figure 1 biomedicines-10-00972-f001:**
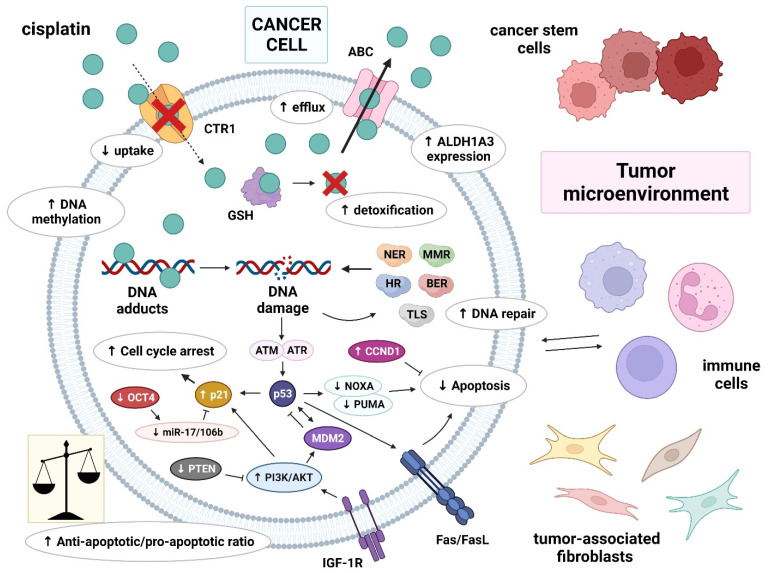
Multiple molecular mechanisms are responsible for cisplatin resistance in germ cell tumors. Cisplatin resistance can be classified as pre-target, on-target and post-target. Pre-target resistance includes: decreased intracellular accumulation of cisplatin (reduced uptake and increased efflux of cisplatin) and increased cisplatin detoxification by cytoplasmic scavengers. On-target resistance refers to an increased ability to repair DNA damage or an acquired ability to tolerate unrepaired DNA lesions. Post-target resistance involves: alterations in apoptosis signaling pathways with a central role of p53, decreased expression of pro-apoptotic factors, and overexpression of anti-apoptotic factors. This leads to cell cycle arrest and the inhibition of apoptosis. Other factors that contribute to cisplatin resistance are: DNA hypermethylation, increased ALDH expression, and a tumor microenvironment with immune cells playing an important role. Abbreviations: ABC = ATP-binding cassette transporter; AKT = protein kinase B (PKB); ALDH1A3 = aldehyde dehydrogenase 1A3; ATM = ataxia telangiectasia mutated; ATR = ATM and RAD3-related; BER = base excision repair; CCND1 = cyclin D1; CTR1 = copper transporter protein; FasL = Fas ligand; GSH = glutathione; HR = homologous recombination; IGF-1R = insulin-like growth factor 1 receptor; miR-17/106b = microRNA-17/106b; MDM2 = mouse double minute 2 homolog; MMR = mismatch repair; NER = nucleotide excision repair; NOXA = Phorbol-12-myristate-13-acetate-induced protein 1; OCT-4 = octamer-binding transcription factor 4; PI3K = phosphoinositide 3-kinase; PTEN = phosphatase and tensin homolog; PUMA = p53 upregulated modulator of apoptosis; TLS = translesion synthesis.

**Table 1 biomedicines-10-00972-t001:** Clinical trials of potential therapeutic agents used in refractory germ cell tumors.

Therapeutic Class/Agent	Therapeutic Target/Mechanism of Action	Patients (N)	Efficacy	Median PFS (CI)	Median OS (CI)	Status	NCT Identifier	Author (Reference)
**Cytotoxic agents**								
Cabazitaxel	Microtubule inhibition	13	CR = 0, PRm+ = 2, SD = 3, PD = 7	7 weeks	23 weeks	Completed		Oing et al., 2020 [131]
		34 (estimated)				Recruiting	NCT02115165	
		29 (estimated)				Recruiting	NCT02478502	
**Tyrosine kinase inhibitors**								
Sunitinib	VEGFR + PDGFR + KIT + RET	10	CR = 0, SD = 5, PD = 5	N/A	N/A	Completed		Feldman et al., 2010 [132]
		33	CR = 0, PR = 3, SD = 13, PD = 15	2.0 months (1.4–2.60)	3.8 months (3.0–6.6)	Completed		Oechsle et al., 2011 [133]
		10	CR = 0, PR = 2	10.8 weeks	12.9 weeks	Completed		Reckova et al., 2012 [134]
		5	CR = 0, PR = 1	N/A	N/A	Completed		Subbiah et al., 2014 [135]
Pazopanib	VEGFR + PDGFR + KIT	43	CR = 0, PR = 2, SD = 19, PD = 16	2.5 months (1.0–3.0)	5.3 months (3.1–15.6)	Completed		Necchi et al., 2017 [136]
Sorafenib	VEGFR + PDGFR + RAF	18	CR/PR = 0, SD = 8	N/A	N/A	Completed		Skoneczna et al., 2014 [137]
Cabozantinib	HGFR (MET) + VEGFR + RET + KIT	25 (estimated)				Recruiting	NCT04876456	
Imatinib	KIT + PDGFR + BCR-ABL	6	CR/PR = 0, SD = 1, PD = 5	N/A	N/A	Completed		Einhorn et al., 2006 [138]
Tivantinib	HGFR (MET)	27	CR/PR = 0, SD = 5, PD = 20	1 month (1–2)	6 months (3–8)	Completed		Feldman et al., 2013 [139]
**mTOR inhibitors**								
Everolimus	mTOR	15	CR/PR = 0, SD = 6	1.7 months (1.1–4.0)	3.6 months (2.0–11.0)	Completed		Mego et al., 2016 [140]
		22	CR/PR = 0, SD = 1, PD = 21	7.4 weeks (4.9–7.6)	8.3 weeks (7.1–9.1)	Completed		Fenner et al., 2019 [141]
Sirolimus + erlotinib	mTOR + EGFR	4 (enrolled)				Terminated (low accrual)	NCT01962896	
**PARP inhibitors**								
Olaparib	PARP	18	CR/PR = 0, SD = 5, PD = 13	N/A	N/A	Completed		De Giorgi et al., 2020 [142]
Veliparib + gemcitabine and carboplatin	PARP + DNMT	15	CR = 0, PR = 4, SD = 5, PD = 6	3.1 months (2.2–3.9)	10.5 months (8.9–11.1)	Completed		Mego et al., 2021 [143]
**CDK inhibitors**								
Palbociclib	CDK 4/6	29 (3 women)	CR/PR = 0, SD = 15	11 weeks	N/A	Completed		Vaughn et al., 2015 [144]
Ribociclib	CDK 4/6	8	CR/PR = 0, SD = 8	N/A	N/A	Completed		Castellano et al., 2019 [145]
**Anti-CD30 agents**								
Brentuximab vedotin	CD30	9	CR = 1, PR = 1, SD = 2, PD = 4	1.5 month (1.4–2.8)	8.0 months (4.6-N/A)	Completed		Necchi et al., 2016 [146]
		5	CR = 1, PR = 1, SD = 1, PD = 2	N/A	N/A	Completed		Albany et al., 2018 [147]
		18 (enrolled)				Terminated (lack of funding/ benefit)	NCT02689219	
**Immune checkpoint inhibitors**								
Pembrolizumab	PD-1	12 (2 women)	CR/PR = 0, SD = 3, PD = 8	2.4 months (1.5–4.5)	10.6 months (4.6–27.1)	Completed		Tsimberidou et al., 2021 [148]
		12	CR/PR = 0, SD = 2, PD = 10	N/A	N/A	Completed		Adra et al., 2018 [149]
Pembrolizumab/Nivolumab	PD-1	7	CR = 0, PR = 1, SD = 1, PD = 2	N/A	N/A	Completed		Zschabitz et al., 2017 [150]
Nivolumab	PD-1	0				Withdrawn	NCT03726281	
Nivolumab + Ipilimumab	PD-1 + CTLA-4	5	CR/PR = 0, SD = 1, PD = 4	N/A	N/A	Completed		McGregor et al., 2020 [151]
		N/A				Recruiting	NCT02834013	
Avelumab	PD-L1	8	PD = 8	0.9 months (0.5–1.9)	2.7 months (1.0–3.3)	Completed		Mego et al., 2019 [152]
Durvalumab +/− Tremelimumab	PD-L1 +/− CTLA-4	22 (11:11)	Combination arm: PR = 1, SD = 1	N/A	N/A	Terminated (loss of accrual)	NCT03081923	Necchi et al., 2018 [153]
Durvalumab + Tremelimumab	PD-L1 + CTLA-4	31 (estimated)				Recruiting	NCT03158064	
**Epigenetic agents**								
Guadecitabine + cisplatin	DNMT	14	CR = 2, PR = 3	1.7 months (0.9–3.7)	7.8 months (2.7, 12.5)	Completed	NCT02429466	Albany et al., 2021 [154]
**Other agents**								
Disulfiram	ALDH	20 (estimated)				Recruiting	NCT03950830	

Abbreviations: ALDH = aldehyde dehydrogenase; BCR-ABL = Philadelphia chromosome; CDK = cyclin-dependent kinase; CI = confidence interval; CR = complete response; CTLA-4 = cytotoxic T-lymphocyte-associated protein 4; DNMT = DNA methyltransferase; EGFR = epidermal growth factor receptor; HGFR = hepatocyte growth factor receptor; KIT = stem cell growth factor receptor (SCFR); mTOR = mammalian target of rapamycin; N = number; N/A = not available; PARP = poly (ADP-ribose) polymerase; PDGFR = platelet-derived growth factor receptor; PD = progressive disease; PD-1 = programmed cell death protein 1; PD-L1 = programmed death-ligand 1; PR = partial response; PRm+ = partial response with positive tumor markers; RAF = rapidly accelerated fibrosarcoma kinase; RET = rearranged during transfection proto-oncogene; SD = stable disease; VEGFR = vascular endothelial growth factor receptor.

## Data Availability

No new data were created or analyzed in this study. Data sharing is not applicable to this article.

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
