# Peer review of "Overcoming Chemotherapy Resistance in Germ Cell Tumors"

_biomedicines, 2022, doi:10.3390/biomedicines10050972_

Round 1

Reviewer 1 Report

Dear Authors,

the topic of each section should be summarized to made them more concise. The Authors should deduce several considerations, reporting the comprehensive data rather than only reporting the results of each literature study. 

Some sentences should be corrected and rephrased to provide a fluid reading and a better understanding (such as lines: 46-49, 52-55, 135-136, 168-170, 264-267, 277-281, 316-319, 346-350, etc.)

The grammatical form should be revised to reduce the repetitions and to provide a better English accuracy.

The caption of Figure 1 should be rephrased and summarized.

Author Response

Dear Reviewer,

thank you for giving us the opportunity to submit a revised draft of our manuscript titled “Overcoming chemotherapy resistance in germ cell tumors” to the Special Issue of Biomedicines: "Killing It Softly–New Approaches to Overcome Cancer Chemoresistance". We appreciate the time and effort that you have dedicated to providing your valuable feedback on our manuscript. We are grateful for your insightful comments on our paper. We have been able to incorporate changes to reflect most of your suggestions. Our revisions are marked in the manuscript file using the “Track changes” function in MS Word. 

Thank you for pointing out the need to revise the grammatical form of the manuscript. Here is a summary of our revisions:

  • We have corrected and rephrased/modified the sentences you mentioned in the review report (lines 46-49, 52-55, 135-136, 168-170, 264-267, 277-281, 316-319, 346-350 and more).
  • We have rephrased and summarized the caption of Figure 1.
  • We have summarized the topic of each chapter in short “Summary” sections.
  • We have corrected the grammatical errors in the manuscript.
  • We have added the content to the section "Future directions and considerations".

Thank you again for taking the time to review our manuscript. We look forward to hearing from you in due time regarding our submission and to respond to any further questions and comments you may have.

Sincerely,

Authors

Reviewer 2 Report

Orszaghova and colleagues propose a review article entitled “Overcoming chemotherapy resistance in germ cell tumors”. Germ cell tumors can be treated with various chemotherapeutics but particularly are sensitive to cisplatin-based chemotherapy. Furthermore, the development of resistance against cisplatin-based chemotherapy leads to disease relapse. Here, the authors discuss various mechanisms leading to the development of drug resistance and discuss various approaches to overcome drug resistance.

The manuscript is straightforward, well written, concise, clear, and lies within the scope of MDPI-Biomedicines.

I am happy to consider this nice review article to be published in the present form.

Author Response

Dear Reviewer,

we would like to thank you for your review report. We appreciate the time and effort that you dedicated to providing your valuable feedback on our manuscript “Overcoming chemotherapy resistance in germ cell tumors”. 

Sincerely,

Zuzana Országhová and colleagues